# Antiobesity and Hypolipidemic Potential of *Nitraria retusa* Extract in Overweight/Obese Women: A Randomized, Double-Blind, Placebo-Controlled Pilot Study

**DOI:** 10.3390/nu16020317

**Published:** 2024-01-21

**Authors:** Aicha Laouani, Hana Nasrallah, Awatef Sassi, Farhana Ferdousi, Feten Zar Kalai, Yosra Hasni, Hiroko Isoda, Saad Saguem

**Affiliations:** 1Laboratory of Metabolic Biophysics and Applied Pharmacology, Faculty of Medicine, University of Sousse, Sousse 4002, Tunisia; laouani_aicha@yahoo.fr (A.L.); hananasrallah.hn@gmail.com (H.N.); awatefsassi@yahoo.com (A.S.); 2USCR Analytical Platform UHPLC-MS & Research in Medicine and Biology, Faculty of Medicine, University of Sousse, Sousse 4023, Tunisia; 3Faculty of Life and Environmental Sciences, University of Tsukuba, Tsukuba 305-8572, Japan; ferdousi.farhana.fn@u.tsukuba.ac.jp; 4Alliance for Research on the Mediterranean and North Africa (ARENA), University of Tsukuba, Tsukuba 305-8572, Japan; zarfeten@gmail.com; 5Japan Laboratory of Aromatic and Medicinal Plants, Center of Biotechnology, Technopark of Borj Cedria, BP 901, Hammam-Lif, Tunis 2050, Tunisia; 6Endocrinology-Diabetology Department, Farhat Hached Hospital, Sousse 4003, Tunisia; yosrahasnielabed@gmail.com; 7Open Innovation Laboratory for Food and Medicinal Resource Engineering (FoodMed-OIL), National Institute of Advanced Industrial Science and Technology (AIST), Tsukuba 305-8577, Japan

**Keywords:** *Nitraria retusa*, obesity, body composition, anthropometric parameters, HDL, TG

## Abstract

This study aimed to assess the efficacy of *Nitraria retusa* extract (NRE) in reducing weight, body mass index (BMI), body fat composition (BF), and anthropometric parameters among overweight/obese women, comparing the results with those of a placebo group. Overweight/obese individuals participated in a 12-week, double-blind, randomized, placebo-controlled trial. Body weight, BMI, body composition, and anthropometric parameters were assessed. Additionally, lipid profile and safety evaluation parameters were evaluated. Compared to the placebo group, the NRE group exhibited a mean weight loss difference of 2.27 kg (*p* < 0.001) at the trial’s conclusion. Interestingly, the most significant weight reduction, amounting to 3.34 kg ± 0.93, was observed in younger participants with a BMI > 30.0. Similarly, BMI and BF% significantly decreased in the NRE group, contrary to the placebo group (*p* = 0.008 and *p* = 0.005, respectively). The percentage of body water (BW) (*p* = 0.006) as well as the ratio of LBM/BF (*p* = 0.039) showed a significant increase after the NRE intervention compared to the placebo. After age adjustment, all variables, except LBM/BF, retained statistical significance. Additionally, all anthropometric parameters were significantly reduced only in the NRE group. Most importantly, a significant reduction in Triglyceride (TG) levels in the NRE group was revealed, in contrast to the placebo group (*p* = 0.011), and the significance was still observed after age adjustment (*p* = 0.016). No side effects or adverse changes in kidney and liver function tests were observed in both groups. In conclusion, NRE demonstrated potent antiobesity effects, suggesting that NRE supplementation may represent an effective alternative for treating obesity compared to antiobesity synthetic drugs.

## 1. Introduction

Obesity (and/or overweight) is defined as excessive fat accumulation in adipose tissue arising when there is an energy surplus between the total amount of calories consumed from food and that spent during metabolic and physical activities [1,2]. It is a multifactorial and growing disease that affects all age ranges in both developed and developing countries [3,4]. Its prevalence has continually increased worldwide over the past few decades [5]. In 2016, 39% of the adults aged over 18 years old in the world were overweight, and 13% were obese [6]. Following this trend, nearly 60% of the world’s adult population is expected to be obese or overweight by 2030 [7]. In Tunisia, 62% of the adult population is overweight, and 28% of the adults are obese [8]. Furthermore, females have almost three times a higher prevalence of obesity than males (13% of obese males and 30% of obese females) [9].

According to the World Health Organization (WHO), the most commonly used parameter for screening obese or overweight subjects is body mass index (BMI), defined as the weight (kg) to height squared (m^2^) ratio [10]. Overweight and obesity are determined according to their BMI values of 25 to 29.9 kg/m^2^ and ≥30 kg/m^2^, respectively [11]. Due to their association with an elevated risk of developing cardiovascular diseases, dyslipidemia, hypertension, and type 2 diabetes, as well as other illnesses, obesity and overweight constitute serious worldwide health problems. The major prejudices to health include a declining quality of life and an increased rate of mortality and morbidity, mostly due to cardiovascular diseases [12,13,14,15,16,17].

Conventional obesity treatment is mainly based on adhering to a strict diet with the practice of regular physical activity and medical or surgical intervention (bariatric surgery) [18]. However, the practice of this invasive procedure should take into account the psychological features to which patients could be exposed after bariatric surgery [19]. Moreover, drugs known to exert an anti-obesity effect could be used in conjunction with dietary changes. Weight-loss medications, including suppressing appetite, reducing food intake, limiting the absorption of food, or altering metabolism, are frequently used [20].

Nevertheless, the intake of such chemical drugs could generate adverse effects, including headache, constipation, nausea, insomnia, increased blood pressure and pulse rate, tachyarrhythmias, and angina pectoris [21].

For all these reasons, research has been focused on the development of new methods as an alternative to conventional anti-obesity treatments that are safe, tolerable, and easy to use, particularly when patients treated with these conventional methods fail to respond to the treatment. One of these new approaches is based on the use of various medicinal plants [22,23,24].

Recently, plants and natural products have been considered promising sources for the discovery of new pharmacological agents to treat obesity and its related complications. It is widely accepted that the compounds involved in managing obesity and its associated diseases are polyphenols, mainly flavonoids and phenolic acids.

These bioactive phytochemicals can work independently or synergistically to increase their effects at different levels of the body. They can modulate lipase activity, appetite, fat synthesis and thermogenesis, satiety, and adipocyte apoptosis. In addition, targeting adipogenesis with various medicinal plants is an important strategy for designing novel anti-obesity drugs that can intervene in adipocyte growth and differentiation with minor adverse effects [3,25,26]. For all these reasons, polyphenols contained in plant extracts have been regarded as a natural therapy that could be potentially used for treating numerous diseases [27], including obesity.

Numerous plants used in traditional medicine, such as *P. vulgaris*, *N. saliva*, *G. pentaphyllum*, *K. carvi*, and *Z. jujuba*, have been proposed to possess beneficial anti-obesity properties without inducing significant side effects [28,29,30]. Therefore, plant extract could be a natural alternative against obesity.”

In this regard, *Nitraria retusa* is a succulent plant belonging to the family *Nitrariaceae*. It is native to areas with high salinities, such as salt marshes and semi-arid saline areas in the deserts of North and East Africa, as well as the Arabian Peninsula and the Middle East [31]. According to previous studies, the beneficial effects of NRE are mainly attributed to the presence of numerous bioactive compounds, including coumarins, alkaloids, phenolic compounds, and especially flavonoids [32,33,34].

Numerous studies have reported that *Nitraria retusa* possesses many biological activities, mainly anticancer [35], antioxidant, antiviral, and antimicrobial properties [36]. In these regards, an in vitro study showed that NRE decreases cell number and size, inhibits differentiation, and reduces fat accumulation and lipid droplet content in 3T3-L1 cells [34]. Moreover, reports have demonstrated the anti-obesity effect of *Nitraria retusa* in obese mice through the regulation of the expression of genes involved in lipolysis and lipogenesis and the enhancement of lipid metabolism in the liver [33,34]. Furthermore, recent observational data showed that daily isorhamnetin, quercetin, and kaempferol consumption is lower in Polish adults showing central obesity than in healthy subjects, suggesting that a diet rich in flavonoids could possibly be protective against central obesity development [37]. Our recent clinical trial has shown the safety and tolerability of *Nitraria retusa* tea in both healthy and overweight/obese individuals for ten days. We further demonstrated its potent hypolipidemic effect through the regulation of HDL and TG after short daily administration intake in overweight and obese participants with a dependent dose effect [38]. These interesting data prompted us to assess the weight-loss-promoting efficacy of *Nitraria retusa* in overweight/obese subjects in a double-blind placebo-controlled study. Thus, this study aimed to evaluate the anti-obesity and hypolipidemic effects of a prolonged daily intake of NRE in women who are overweight or obese following a 12-week intake.

## 2. Materials and Methods

### 2.1. Study Design

This clinical pilot study spanning 12 weeks was conducted to evaluate the effect of dietary supplementation with NRE on diverse parameters, including body weight, body composition, anthropometric measurements, clinical indicators, hematological parameters, and biochemical factors in overweight and obese adults. Employing a randomized, placebo-controlled, double-blind, parallel-group design, the study was conducted at the Laboratory of Biophysics in the Faculty of Medicine, University of Sousse, Tunisia, and the Department of Endocrinology and Diabetology, Farhat-Hached University Hospital, Sousse, Tunisia. Recruitment of participants from the Sousse area occurred through advertisements from May 2021 to August 2023. Volunteers from various regions of Tunisia meeting inclusion/exclusion criteria were included in this study. The study subjects signed a written informed consent to participate in the clinical trial.

### 2.2. Study Participants

A total of 120 participants were recruited. Among them, 35 were unlikely to meet the inclusion criteria. Then, a total of 85 participants were invited to evaluate the eligibility criteria. Among them, 17 participants did not meet the inclusion criteria, nine declined to participate for personal reasons, and an additional eight did not confirm their availability for the baseline visit. Ultimately, 68 participants were randomized and assigned to receive either NRE (*n* = 38) or placebo extract (*n* = 30) once daily (after dinner). Finally, 30 participants in the NRE group and 13 in the placebo group completed all aspects of the study and were the primary focus of analysis.

Inclusion criteria: Overweight and obese females aged between 20 and 75 years and with a BMI > 25 kg/m^2^ were eligible for this study.

“Although our primary consideration was an equitable representation of age to ensure a balanced demographic, we had a higher average age in the placebo group than the NRE group.”

Exclusion criteria: Individuals with, or suspected of having, medical conditions that could impede their involvement in the trial were excluded from participation. These conditions included metabolic pathologies, diabetes, thyroid disorders, liver and kidney diseases, as well as heart disease or severe hypertension. Furthermore, heavy drinkers and smokers, individuals who had undergone bariatric surgery, and subjects using weight loss medication or diet supplements were excluded. Furthermore, individuals using corticoid treatments during the three months preceding the inclusion were also excluded. Pregnant women and women using contraceptives who had experienced substantial alterations in dose or type within the three months leading up to their inclusion were also excluded. Women experiencing menopause during the study period were excluded.

Finally, individuals who had participated in other clinical trials within the three months preceding the current trial were not eligible for inclusion.

### 2.3. Preparation of NRE and Placebo

*Nitraria retusa* was harvested from saline soils in Kairouan, mid-Tunisia, on 10 April 2021. Leaves were rinsed with water, air-dried in the shade, and at room temperature for 15 days, and then ground to a fine powder using a Moulinex mill-type device (MOULINEX SA, Paris, France).

The NRE and placebo were provided to the study participants in sachets containing 2000 mg of the powder, and they were instructed on how to prepare the daily infusion. The NRE was prepared by dissolving 2000 mg of crushed dried leaf powder of *Nitraria retusa* in 100 mL of boiling water for 15 min.

Qualitative and quantitative analyses of NRE were performed using a confirmed high-performance liquid chromatography (HPLC) method (UHPLC-DAD, SHIMADZU 8045). The administered dose, containing 20 mg of flavonoids and 16 µg of isorhamnetin, was selected as previously described [38].

The placebo extract was formulated by dissolving 2000 mg of an herbal mixture powder, identical in appearance to NRE, in 100 mL of boiling water. This herbal mixture does not contain any active substances and has been verified to be safe for human consumption.

### 2.4. Study Protocol

During the screening visit, nutritional and clinical histories were recorded, and consent forms were signed. Anthropometric, biochemical, and hematological parameters were evaluated. The 85 subjects were then enrolled in a 12-week intervention; among them, 68 participants were randomized and assigned to receive either NRE or placebo extract once daily (after dinner). Clinical, anthropometric, and body composition parameters were measured, and blood samples were collected for biochemical and hematological analysis at baseline and after 12 weeks. Adverse side events and associated medications were also recorded. Regular monitoring was carried out throughout the study period.

### 2.5. Nutritional Consultation

The selected subjects received a consultation from a nutritionist experienced in treating obese and overweight participants at baseline entry into the trial and at the end of the study after 12 weeks. Participants were instructed not to make any important changes to their diet, routine, or lifestyle habits during the 12-week trial period.

All participants filled out a questionnaire, providing details regarding nutritional status and medical history at baseline and after the 12-week study period.

### 2.6. Primary/Secondary Outcomes

The primary outcome of this study was the efficacy of NRE intake on weight loss. The secondary outcomes were the improvement of anthropometric measurements of obese subjects, body composition, lipid profiles, and safety monitoring tests. Moreover, adverse side effects that could be associated with the treatments were identified by a medical expert. All measurements were evaluated early in the morning with an empty stomach at baseline and at the end of the follow-up. Blood samples for biochemical analysis were collected following a 12 h overnight fast at baseline and at week 12.

#### 2.6.1. Primary Outcome

##### Body Weight Measurements

Participants were weighted without shoes and in light clothing using a bioelectrical impedance analysis (BIA) machine within 0.1 kg intervals (Beurer digital diagnostic scale, Model BG63, Ulm, Germany).

#### 2.6.2. Secondary Outcomes

##### Anthropometric Measurements

Subjects were monitored for anthropometric parameters, counting waist circumference (WC) and hip circumference (HC) to the nearest 0.1 cm using non-stretchable plastic tape. Height was measured with a wall-mounted Stadiometer. The WC was measured by placing the plastic tape at the umbilicus point, and the HC was measured in cm over the buttocks. The waist-to-hip ratio (WHR) was then calculated by dividing the waist and hip circumferences. BMI (kg/m^2^) was calculated based on the following formula: BMI = weight/height^2^.

The mid-thigh measurement is a circumference measurement around the middle of the thigh, from the inguinal crease to the proximal border of the patella. Half an arm was measured in cm, with the subject standing upright with the back straight.

##### Body Composition Measurements with Bioelectrical Impedance (BIA)

Body composition, including body fat % (BF), body water % (BW), and lean body mass % (LBM), was measured without shoes by a bioelectrical impedance analysis (BIA) machine (Beurer digital diagnostic scale, Model BG63, Ulm, Germany).

##### Lipid Blood Profile Analysis

Lipid profile parameters, including triglycerides (TG), total cholesterol (TC), and high-density lipoprotein (HDL) cholesterol, were assessed by spectrophotometric technique using the Beckman Coulter D600 analyzer (Brea, California, US). Low-density lipoprotein (LDL) cholesterol was then calculated using the Friedewald formula: LDL = TC-HDL − TG/5.

##### Safety Analysis and Assessment

Renal function markers include creatinine (s Cr) and urea (BUN), otr’and liver function tests such as alanine transaminase (ALT), aspartate transaminase (AST), total bilirubin (TB), direct bilirubin (DB), albumin (Alb), gamma glutamyl transferase (GGT), and alkaline phosphatase. (ALP) were measured by spectrophotometric technique using a Beckman Coulter D600 analyzer.

Hematological parameters (white blood cells (WBC), red blood cells (RBC), hemoglobin (Hb), hematocrit (Ht), and platelet (Plt)) were assessed using a Celltac alpha hematology analyzer.

Vital signs, including systolic blood pressure (SBP), diastolic blood pressure (DBP), and pulse rate, were carried out by medical staff using a standard calibrated mercury sphygmomanometer after the participants had been rested for at least 15 min.

### 2.7. Compliance and Adverse Events

A participant was considered compliant when consuming the contents of ≥95% of the treatment, an adverse event, as any unfavorable, accidental effects along with other symptoms involving vomiting, nausea, diarrhea, constipation, and halitosis reported by participants or observed by the investigator were recorded during the experimental periods. Subjects were informed of their right to withdraw from the study at any time.

This study supplements were well tolerated, and no symptoms of intolerance were declared. Yet, periodic laxative and diuretic effects lasting for three days were the most common events occurring during the trial.

### 2.8. Ethical Statement

This study was conducted according to the guidelines of the Declaration of Helsinki and approved by the Human Research Ethics Committee of the Faculty of Medicine of Sousse, Tunisia (CEFMS 34/2019, 18 November 2019).

### 2.9. Statistical Analysis

Statistical analyses were performed using SPSS 28.0 for Windows (IBM Corp., Armonk, NY, USA). Data are expressed as the mean ± standard deviation (SD). The normality of the variables was confirmed using the Shapiro–Wilk test. The homogeneity of the variance was assessed using Levene’s test.

Changes in the primary and secondary outcomes from baseline to week 12 were defined by the absolute difference of the value of a parameter in week 12 minus the value at baseline. Within-group differences at baseline and after 12 weeks were compared using the paired *t*-test for normally distributed data and the Wilcoxon Signed Rank Test for nonparametric data when normality was rejected.

Between-group differences (Placebo vs. NRE intervention) were evaluated by an independent sample *t*-test for parametric and equal variances and Welch’s *t*-test for unequal variances for normal data distribution. The statistical analysis for nonparametric data were assessed by the Mann–Whitney U test. The results were considered statistically significant at *p* < 0.05. Finally, as age was significantly different at baseline between the treatment groups, we applied age adjustment using the parametric ANCOVA or Quade’s nonparametric ANCOVA to control the effects of age on the primary and secondary outcome variables.

## 3. Results

### 3.1. Study Flow and Participant Characteristics

Among the 120 recruited participants, 35 did not meet the inclusion criteria and were excluded. A total of 85 female subjects were examined during the screening visit. After completing the selection process, 17 were excluded: 9 suitable subjects declined to participate, and 8 refused to take part in the trial for personal reasons. A total of 68 overweight and obese subjects were included in this study and were randomly distributed into NR and placebo groups. Approximately 38 participants were allocated to the NR group, and among them, 30 completed the trial. While 30 participants were assigned to the placebo group, among them, 13 completed the study period. Participants who consumed NRE experienced an improvement in sleep, a reduced appetite, anti-bloating, and diuretic effects. Moreover, they noticed a change in body silhouette during the study period, which might have motivated the participants to adhere to the intervention. However, participants in the placebo groups did not perceive any improved outcomes, and no effects were reported, which may explain the high dropout rate of the placebo group compared to the NRE group. The diagram of the study design for the present clinical trial is provided in Figure 1.

The demographic data and baseline characteristics of the subjects are summarized in Table 1.

No significant differences in all baseline measurements were observed except for age, which was higher in the placebo group compared to the NRE group (*p* = 0.034) (Table 1).

About 74% of the participants (*n* = 32) were obese, and 26% (*n* = 11) were overweight.

### 3.2. Effect of NRE Supplementation on Body Weight, Changes in Body Composition, and Anthropometric Parameters

Changes in variables over the 12-week treatment for the placebo and NRE groups are shown in Table 2.

The NRE group showed a significant weight reduction after 12 weeks of intervention (*p* < 0.001), whereas it was slightly increased in the placebo group (*p* = 0.83). The between-group comparison revealed a significant weight loss (*p* <0.001). Therefore, NRE has a positive effect on lowering body weight. Similarly, BMI was significantly decreased (*p* < 0.001) in the NRE group, contrary to the placebo group (*p* > 0.05). The mean BMI loss between the NRE and placebo groups was significant (*p* = 0.008). A significant %BF loss was only observed within the NRE group after 12 weeks of treatment (*p* = 0.001). The difference between the placebo group and the NRE group was significant (*p* = 0.005).

The percentages of LBM and BW displayed significant increases after NRE intervention (*p* = 0.045 and *p* < 0.001, respectively) compared to the placebo group. The between-group comparison revealed a significant increase in BW % (*p* = 0.006) (Figure 2).

In addition, the WC and WHR, mid-thigh, and half arm were reduced significantly only in the NRE group (Figure 3).

After age adjustment, weight, BMI, BF%, BW%, WC, and MTC parameters showed significant differences.

Our results suggest a beneficial effect of NRE on body composition and on anthropometric parameters during 12 weeks of intake.

### 3.3. Effect of NRE Supplementation on Lipid Profile

As shown in Table 3, NRE administration revealed a significant reduction in TG levels (*p* = 0.011). The significance was observed even after age adjustment (*p* = 0.016).

Although not statistically significant, an improvement in HDL levels was observed in the NRE group. The placebo group, on the other hand, displayed an elevation in HDL and TG levels after the 12-week treatment (Figure 4). No difference was observed regarding LDL and TC levels in both groups.

Taken together, our results suggest that NRE might have a beneficial effect on lipid profiles in obese subjects.

### 3.4. Effect of BMI and Age on Responses to Interventions

We additionally performed stratified analyses (as detailed in Appendix A) to examine the effect of BMI and age on body weight in the NRE group compared to placebo participants.

Our results indicated that among participants with a BMI less than 29.9, weight loss was 1.01 kg ± 0.67 compared to an increase of 0.83 kg ± 1.19 in the placebo group (*p* = 0.048). (Appendix A). Nevertheless, among participants with a BMI exceeding 30.0, a better weight reduction was observed (2.86 kg ± 0.61) compared to the placebo group (*p* < 0.001) (Appendix A). In addition, a younger subject, under 40 years old, exhibited a weight decrease of 2.51 kg ±0.71 compared to an increase of 1.05 kg ± 0.79 for the placebo group (*p* = 0.003) (Appendix A). On the other hand, subjects above 40 years old showed a 1.48 kg ± 0.39 weight loss compared to 0.42 kg ± 0.45 for the placebo group (*p* = 0.01) (Appendix A). Moreover, individuals with a BMI < 30.0 and aged below 40 years showed a body weight loss of −1.13 ± 0.6 compared to an increase of +1.3 ± 1.9 in the placebo group (Appendix A). Interestingly, the highest weight decrease of 3.34 kg ± 0.93 was observed in younger participants with a BMI > 30.0 (Appendix A), compared to 2.06 ± 0.29 in the oldest group (Appendix A). Hence, a follow-up study specifically targeting obese women younger than 40 years of age could provide a more interesting weight loss study, allowing the evaluation of the real antiobesity potential of NRE.

### 3.5. Safety Issues and Adverse Events

To evaluate the safety of the intervention, hemodynamic and hematological parameters and kidney and liver function tests were measured in both groups. During the screening phase for study enrollment, all participants exhibited normal blood biochemical values. The results are presented in Table 4.

Based on our results, no significant differences were observed in SBP, DBP, or pulse rate within or between the two groups during the 12 weeks study period.

Furthermore, there were no noticeable changes in hematological parameters, including WBC count, RBC count, Hb, Ht, and Plt count, between both placebo groups and NRE for 12 weeks.

Concerning renal function and liver function tests, no significant differences were observed at the end of the trial compared to the baseline except for the ALP levels, which were higher in the NRE group after 12 weeks of intervention. Yet, values were within normal limits.

Our results suggested the safety of daily consumption of NRE in terms of kidney and liver functions and hematological and clinical parameters, encompassing overweight and obese participants. Both placebo and NRE infusions were well-tolerated. However, during the first three-four days of the study period, few participants from NRE stated a diuretic or laxative effect, but no important adverse events were reported during the physical examinations. It is important to note that participants in the NRE group noticed an improvement in sleep during the study period.

## 4. Discussion

To the best of our knowledge, this is the first clinical trial evaluating the effects of NRE intake on body weight, anthropometric parameters, body composition, biochemical, hematological, and clinical parameters in overweight/obese adult women compared to placebo-supplemented subjects during a twelve-week intervention. Our results showed that the consumption of 2000 mg/day of *Nitraria retusa* significantly decreased body weight, BMI, and WC and improved fat-related parameters and lipid profiles.

Our findings illustrate the anti-obesity properties of NRE. Indeed, the daily intake of NRE over three months induced in obese and overweight subjects significant decreases in weight and fat body as well as in BMI. Elsewhere, both BMI and body weight are found to be closely correlated with BF%, suggesting that the significant body loss observed could be related to the decrease in BF%. The significant loss of BF% was similarly observed, along with a significant improvement in BW%, in LBM%, and in the LBM-to-BF ratio, in the NRE group compared to the placebo group. These results indicate that NRE intake may potentially control BF accumulation and weight and would help people lose weight from the fat compartments of the body. The relationship between BF and BW is inverse. As previously mentioned, fat tissues contain less water than lean tissues. A lean body mass contains 70–75% water, while fat tissues contain only about 10% water [39]. Moreover, intracellular water content in lean mass has been proposed as an indicator of cell hydration and muscle quality [40]. The reduction observed in our study in total body weight with BF% and WC was balanced by increases in BW%, which ameliorates LBM% [41,42].

Our results are in accordance with a previous study that proposed that NRE treatment had an anti-obesity effect through the modulation of the expression of genes involved in lipogenesis and lipolysis in high-fat diet-induced obesity in mice [34]. Another study suggested that NRE decreases body weight and fat mass in obese mice for a period of 4 weeks [33].

In addition, waist circumference measurements are widely used as a tool for evaluating the anti-obesity effect induced by the intake of medicinal plant extracts. In fact, higher fat distribution over the abdominal region is associated with increased WC. Accordingly, WC reflects the level of abdominal fat mass and represents mostly visceral adipose [43,44].

In this study, the intake of a daily NRE over three months was found to display significant decreases in WC values. The reduced WC may also be due to the laxative and anti-bloating effects of NRE reported during the study period. NRE may stimulate gastric secretion and intestinal peristalsis and normalize the flora in the colon, thus accelerating intestinal transit.

In addition, the results stratified by age and BMI suggest that obese women under the age of 40 respond particularly to NRE consumption and showed greater weight loss than seen in our total population. Our results suggest that NRE is more effective on young and obese women compared to older women and represents a promising source of a new natural anti-obesity drug.

Therefore, in addition to its beneficial effects against obesity, the daily intake of NRE showed hypolipidemic effects compared to a placebo intervention. This implies that the weight loss, favorable changes in body composition, anthropometric parameters, and lipid profile were probably associated with the bioactive components in NRE. In fact, the bioactive compounds of NRE were widely studied for their anti-obesity properties. Numerous studies have demonstrated that NRE contains significant quantities of flavonoids such as quercetin, isorhamnetin, kaempferol, apigenin, luteolin, and others that work in synergy and may be responsible for its beneficial effects [33,34]. Studies have demonstrated that these compounds may inhibit adipogenesis in 3T3-L1 cells and reduce body fat accumulation and weight in a mouse model of HFD-induced obesity [25,45,46,47].

Apigenin has been shown to reduce adipogenesis in 3T3-L1 cells through the activation of AMPK and the decrease in adipogenic and lipolytic gene expression [47].

Another study showed that luteolin reduced adiposity through the up-regulation of the expression of genes controlling lipolysis and enhancing fatty acid uptake genes in mice with diet-induced obesity for 16 weeks [46].

Moreover, quercetin and kaempferol were found to be effective in inhibiting the differentiation of preadipocyte 3T3-L1 cells into mature adipocytes by down-regulating genes involved in adipocyte differentiation [48]. In vivo data revealed that kaempferol and quercetin showed a marked reduction in serum TG, TC, and LDL levels in HFD- stitozotocin-induced diabetic mice [48].

Ganbold et al. (2019) assessed the effects of isorhamnetin in murine 3T3-L1 preadipocytes for a 14-day incubation period and demonstrated a decrease in lipid droplet formation, indicating adipogenesis inhibition [45]. Another study suggested that isorhamnetin incubation with preadipocytes for nine days prevents lipid accumulation and inhibits adipocyte differentiation via the regulation of genes implicated in lipid metabolism [2]. Studies using animal models have also revealed the potential anti-obesity effects of isorhamnetin and isorhamnetin glycosides. Indeed, its beneficial effects on body weight reduction, adipocyte size, and energy metabolism were demonstrated in mice treated for a long period of time (4 or 12 weeks) [49,50]. Isorhamnetin treatment revealed a significant TG and TC decrease with a significant elevation of HDL in diabetic rats [51]. At the molecular level, isorhamnetin could regulate the expression of the major adipocyte markers, mainly the transcription factors PPAR γ and CCAAT/enhancer-binding protein-α (C/EBPα), one of the main regulators of adipogenesis and cell differentiation [52,53]. Importantly, a recent study revealed that isorhamnetin could significantly reduce fat amounts in the animal body via the PPAR-α-dependent pathway [54].

Recent observational data showed that daily isorhamnetin, quercetin, and kaempferol consumption is lower in Polish adults showing central obesity than in healthy subjects, suggesting that a diet rich in flavonoids could possibly be protective against central obesity development [37]. Therefore, NRE intake for 12 weeks could prevent adipogenesis and modulate genes implicated in lipid metabolism.

All these observations support the potential beneficial effects of the NRE against obesity and associated cardiovascular risk factors and suggest a plausible phytotherapeutic approach for the use of the NRE in the management of obesity as an alternative to synthetic anti-obesity drugs.

Some limitations of this study should be acknowledged. Firstly, this trial was conducted with overweight/obese women in order to have a homogenous population. Thus, studies on the anti-obesity properties of *Nitraria retusa* in men and children with a larger number of participants are proposed. Secondly, in our study, overweight/obese individuals with other medical disorders, such as metabolic complications and cardiovascular disease, were excluded. Therefore, studies examining the anti-obesity properties of NRE in participants with obesity-associated complications are suggested.

## 5. Conclusions

This study demonstrated, for the first time, that 12-week NRE consumption (2000 mg/day) may reduce body weight, WC, and BMI, mainly explained by body fat loss balanced with an increase in body water as well as muscle mass. Those changes are accompanied by an improvement in the lipid profile implicated in the prevention of cardiovascular diseases and obesity-related complications. These findings suggest a plausible phytotherapeutic approach for the use of NRE in the management of obesity as an alternative to synthetic anti-obesity drugs. Nevertheless, the molecular mechanism of action of the active principle(s) of NRE remains to be determined. Our future plan is to identify and quantify the different phenolic compounds of *Nitraria retusa* aqueous extract, mainly the glycosylated flavonoids of isorhamnetin. Molecular docking will also be used to further understand the binding interaction and affinity of the identified molecules towards the human PPARα receptor, which controls dyslipidemia and regulates lipid catabolism and inflammation.

## Figures and Tables

**Figure 1 nutrients-16-00317-f001:**
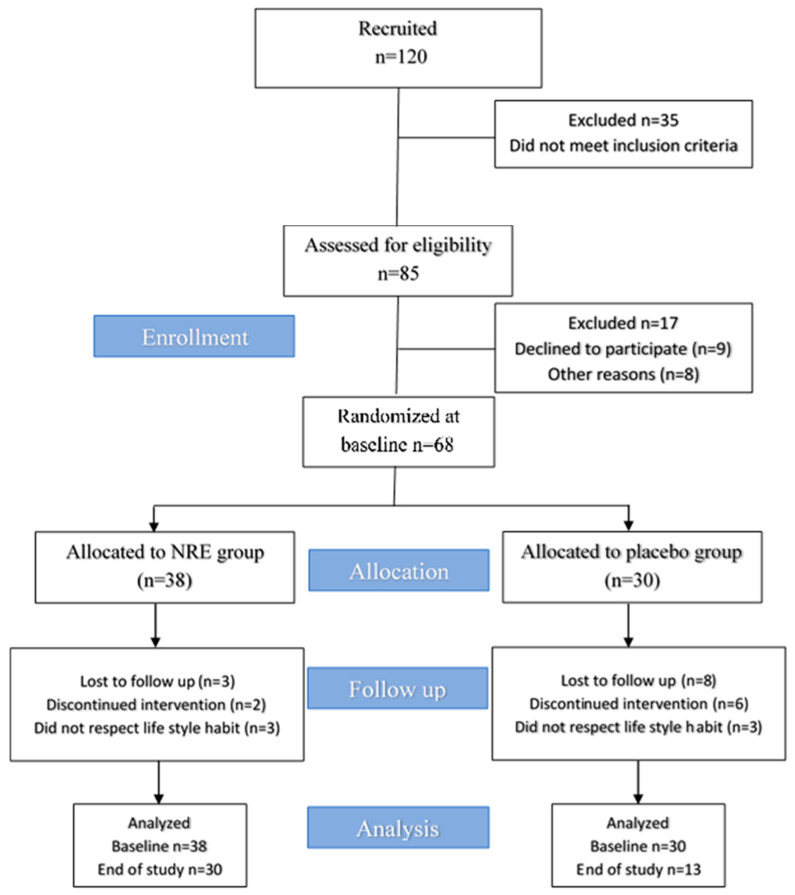
Trial flow chart of this study.

**Figure 2 nutrients-16-00317-f002:**
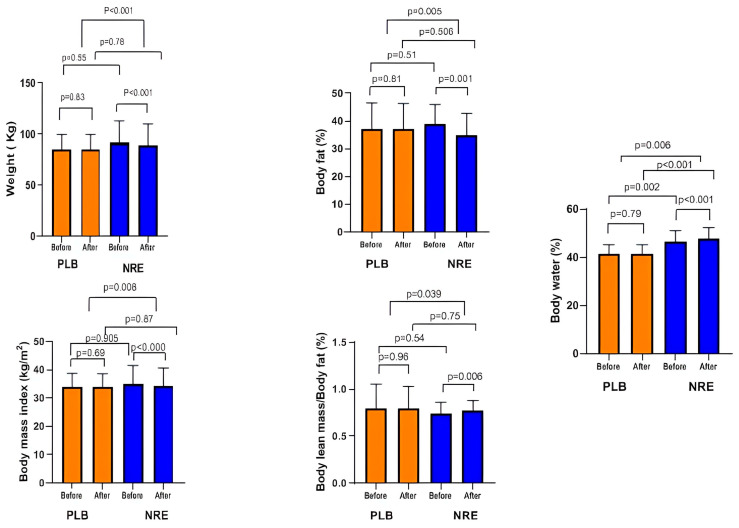
Body weight and body composition parameters of the overweight/obese study participants before and after 12-week *Nitraria retusa* intake compared to placebo group.

**Figure 3 nutrients-16-00317-f003:**
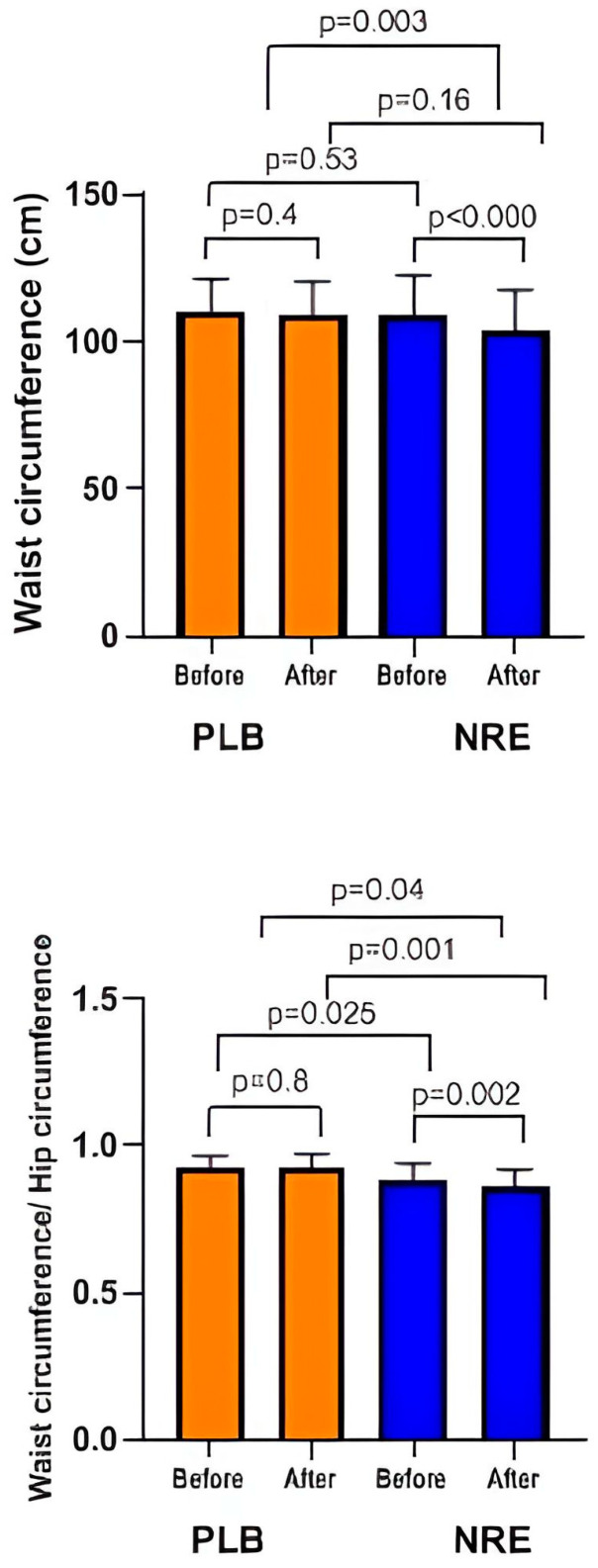
Waist circumference and waist circumference/hip circumference ratio of the overweight/obese study participants before and after 12-week *Nitraria retusa* intake compared to placebo group.

**Figure 4 nutrients-16-00317-f004:**
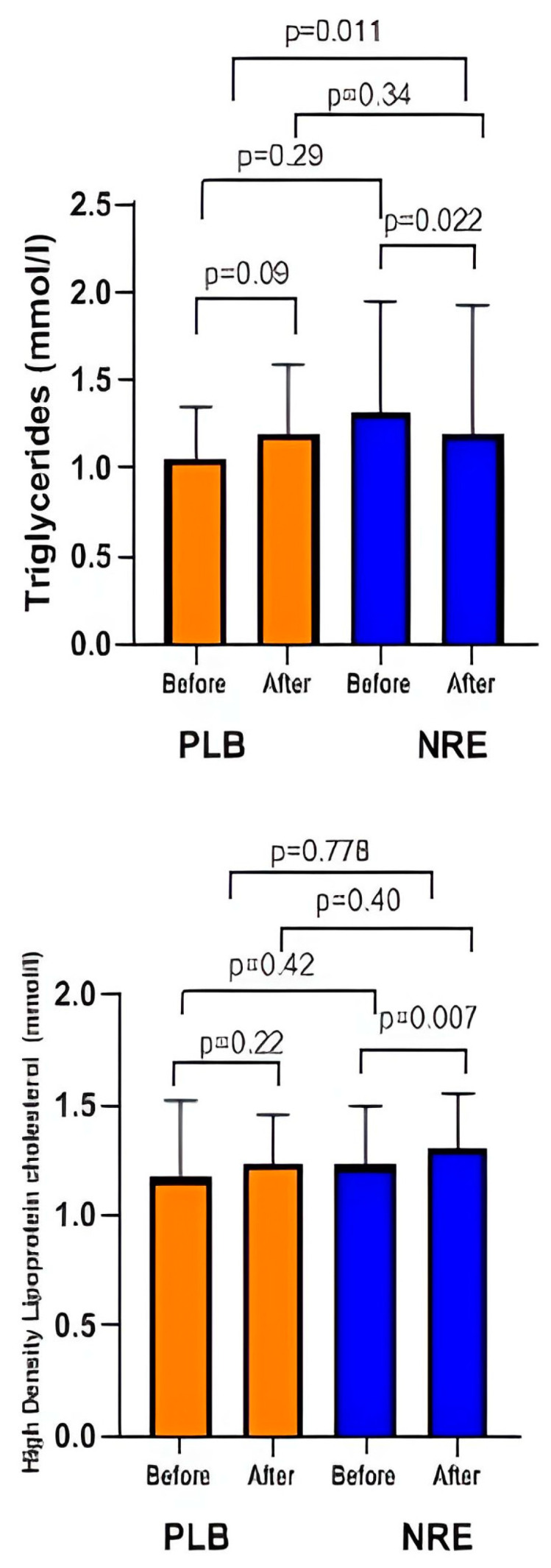
Lipid profile parameters before and after 12 weeks of NRE intervention in overweight and obese participants compared to placebo participants.

**Table 1 nutrients-16-00317-t001:** Baseline characteristics of the overweight/obese study participants.

	Placebo Group (*n* = 13) 13 Women		NRE Group (*n* = 30) 30 Women		*p* Value ^a^
Variables	Mean	(SD)	Mean	(SD)	
Age, y (range)	45.92	(33–59)	39.33	(24–59)	0.034
Height, cm	158.08	(1.79)	161.13	(1.16)	0.55
Weight, kg	84.83	(4.11)	91.43	(3.93)	0.322
BMI, kg/m^2^	33.86	(1.33)	34.84	(1.22)	0.905

^a^ Between-group differences (NRE group vs. Placebo group) were measured by an unpaired *t*-test (parametric) and a Mann–Whitney U test (nonparametric) for continuous variables.

**Table 2 nutrients-16-00317-t002:** Body composition and anthropometric parameters of the overweight/obese study participants before and after 12-week *Nitraria retusa* intake compared to placebo group.

Variables (Units)		Placebo Group ^a^ (*n* = 13)				NRE Group ^a^ (*n* = 30)			Between Group		
	Baseline	After 12 Weeks	Δ ^b^	*p*-Value ^c^	Baseline	After 12 Weeks	Δ ^b^	*p*-Value ^c^	Δ ^d^	*p*-Value ^e^	Adjusted *p*-Value ^f^
Body composition											
Weight (kg)	84.83 (4.11)	84.91 (4.13)	+0.08 (0.39)	0.835	91.47 (3.9)	89.20 (3.81)	−2.27 (0.31)	<0.001	−2.35 (0.08)	<0.001	<0.001
BMI (kg/m^2^)	33.86 (1.33)	33.78 (1.33)	−0.076 (0.191)	0.696	34.84 (1.22)	34.08 (1.19)	−0.76 (0.139)	<0.001	−0.684 (0.052)	0.008	0.006
BF (%)	36.95 (2.72)	36.95 (2.72)	+0.028 (0.23)	0.811	40.31 (1.03)	39.39 (0.91)	−0.92 (0.24)	0.001	−0.948 (0.01)	0.005	0.047
BLM (%)	27.28 (0.71)	27.40 (0.63)	+0.11 (0.17)	0.676	29.62 (0.54)	29.9 (0.53)	+0.27 (0.125)	0.045	0.259 (0.05)	0.058	0.071
BLM/BF	0.79 (0.007)	0.79 (0.06)	−0.0025 (0.009)	0.960	0.74 (0.03)	0.76 (0.029)	+0.023 (0.006)	0.006	0.0255 (0.054)	0.039	0.056
BW (%)	41.50 (1.10)	41.55 (1.08)	+0.05 (0.18)	0.794	44.05 (0.88)	44.83 (0.76)	+0.78 (0.18)	<0.001	0.73 (0.001)	0.006	0.005
Anthropometric parameters											
WC (cm)	109.8 (3.2)	108.88 (3.2)	−0.92 (1.06)	0.405	108.75 (2.52)	104.25 (2.45)	−4.5 (0.59)	<0.001	−3.58 (0.47)	0.003	0.012
WHR	0.926 (0.11)	0.928 (0.121)	+0.002 (0.11)	0.805	0.886 (0.01)	0.86 (0.01)	−0.21 (0.006)	0.002	−0.212 (0.104)	0.04	0.056
Mid-thigh (cm)	57.23 (1.13)	57 (1.02)	−0.23 (0.68)	0.741	60.05 (1.31)	57.31 (1.27)	−2.73 (0.45)	0.002	−2.5 (0.23)	0.004	0.004
Half arm (cm)	36.92 (1.3)	36.30 (1.43)	−0.61 (0.72)	0.416	35.93 (1.04)	33.49 (1.48)	−2.43 (1.2)	<0.001	−1.82 (0.48)	0.51	0.240

^a^ Data are expressed as the mean (SD). ^b^ Mean difference at baseline and after 12 weeks of intake (12 weeks − baseline). ^c^ Within-group differences before and after intervention were compared using a paired *t*-test (parametric) and a Wilcoxon Signed Ranks test (nonparametric). ^d^ Mean difference between groups (NRE vs. Placebo) at 12 weeks. ^e^ Mean difference at 12 weeks was compared between groups by independent samples *t*-tests (parametric and equal variances), Welch’s *t*-test (unequal variances), and Mann–Whitney U test (nonparametric). ^f^ Adjusted *p*-value was calculated using ANCOVA, or Quade’s nonparametric ANCOVA, adjusted for age.

**Table 3 nutrients-16-00317-t003:** Lipid profile of the overweight/obese study participants after 12 weeks *Nitraria retusa* intake compared to placebo group.

Variables (Units)	Placebo Group ^a^ (n = 13)				NRE Group ^a^ (n = 30)				Between Group		
	Baseline	After 12 Weeks	Δ ^b^	*p*-Value ^c^	Baseline	After 12 Weeks	Δ ^b^	*p*-Value ^c^	Δ ^d^	*p* Value ^e^	Adjusted *p*-Value ^f^
Lipid profile											
HDL, mmol/L	1.17 (0.096)	1.228 (0.06)	+0.057 (0.07)	0.224	1.22 (0.04)	1.29 (0.046)	+0.07 (0.02)	0.007	0.013 (0.05)	0.778	0.985
TG, mmol/L	1.056 (0.081)	1.19 (0.111)	+0.13 (0.07)	0.095	1.32 (0.11)	1.19 (0.13)	−0.127 (0.056)	0.022	−0.257 (0.014)	0.011	0.016
LDL, mmol/L	3.30 (0.211)	3.29 (0.17)	−0.006 (0.16)	0.968	3.28 (0.12)	3.25 (0.14)	−0.027 (0.10)	0.805	−0.021 (0.06)	0.92	0.875
TC, mmol/L	4.93 (0.260)	5.03 (0.197)	+0.09 (0.20)	0.648	5.11 (0.182)	5.12 (0.19)	+0.1 (0.09)	0.902	0.01 (0.11)	0.68	0.558

^a^ Data are expressed as the mean (SD). ^b^ Mean difference at baseline and after 12 weeks of intake (12 weeks − baseline). ^c^ Within-group differences before and after intervention were compared using a paired *t*-test (parametric) and a Wilcoxon Signed Ranks test (nonparametric). ^d^ Mean difference between groups (NRE vs. Placebo) at 12 weeks. ^e^ Mean difference at 12 weeks was compared between groups by independent samples *t*-tests (parametric and equal variances), Welch’s *t*-test (unequal variances), and Mann–Whitney U test (nonparametric). ^f^ Adjusted *p*-value was calculated using ANCOVA, or Quade’s nonparametric ANCOVA, adjusted for age.

**Table 4 nutrients-16-00317-t004:** Clinical, hematological, and biochemical parameters of the overweight/obese study participants before and after 12 weeks *Nitraria retusa* intake compared to placebo group.

Variables (Units)	Placebo Group ^a^ (*n* = 13)				NRE Group ^a^ (*n* = 30)				Between Groups	
	Baseline	After 12 Weeks	Δ ^b^	*p*-Value ^c^	Baseline	After 12 Weeks	Δ ^b^	*p*-Value ^c^	Δ ^d^	*p*-Value ^d^
Clinical parameters										
SBP, mmHg	122.31 (3.78)	125.38 (3.511)	3.07 (2.08)	0.157	117.67 (2.97)	119.78 (2.93)	2.11 (1.85)	0.860	−0.96 (0.23)	0.176
DBP, mmHg	73.08 (3.028)	77.69 (3.028)	4.61 (6.16)	0.468	69.33 (1.97)	72.67 (1.72)	3.33 (2.1)	0.105	−1.28 (4.06)	0.98
Pulse rate	73.08 (1.98)	71.92 (2.49)	−1.15 (2.05)	0.586	79.03 (1.76)	79.77 (1.59)	0.73 (1.54)	0.638	1.88 (0.51)	0.49
Hematological parameters										
RBC, 10^12^/L	4.40 (0.117)	4.298 (0.111)	−1.02 (0.09)	0.289	4.35 (0.07)	4.4 (0.09)	0.054 (0.06)	0.387	1.074 (0.03)	0.169
WBC, 10^9^/L	6.762 (0.504)	6.077 (0.415)	−0.68 (0.54)	0.230	6.45 (0.28)	6.57 (0.275)	0.123 (0.149)	0.415	0.803 (0.39)	0.172
Hb, gm/dL	12.354 (0.362)	12.269 (0.369)	−0.08 (0.25)	0.748	13.2 (0.77)	12.18 (0.26)	−1.02 (0.70)	0.115	−0.94 (0.45)	0.39
Ht, %	36.4 (0.975)	35.85 (0.958)	−0.54 (0.72)	0.465	36.76 (0.7)	37.45 (0.68)	0.69 (0.51)	0.553	1.23 (0.21)	0.8
PLT, 10^9^/L	312.62 (20.44)	277.54 (25.88)	−35.07 (32.84)	0.307	268.83 (10.5)	263.6 (10.6)	−5.23 (8.87)	0.560	29.84 (23.97)	0.195
Liver Function Test										
ALT, UI/L	10.05 (1.19)	10.08 (0.79)	0.03 (1.18)	0.89	10.4 (1.01)	11.93 (1.24)	1.53 (0.93)	0.67	1.5 (0.25)	0.51
AST, UI/L	18.15 (2.41)	20.85 (0.629)	2.69 (2.32)	0.269	19.43 (0.81)	19.8 (0.988)	0.36 (1.15)	0.623	−2.33 (1.17)	0.32
ALP, UI/L	64.69 (5.64)	73.92 (5.50)	9.23 (4.41)	0.058	57.17 (2.53)	60.47 (2.86)	3.3 (1.44)	0.166	−5.93 (2.97)	0.314
GGT, UI/L	22.38 (2.94)	21.15 (0.40)	−1.23 (2.18)	0.40	19.57 (2.84)	21.37 (2.87)	1.8 (1.69)	0.12	3.03 (0.49)	0.312
BT, μmol/L	4.92 (0.431)	5.38 (0.488)	0.46 (0.75)	0.645	6.5 (0.649)	6.8 (0.63)	0.3 (0.55)	0.035	−0.16 (0.2)	0.95
BD, μmol/L	1.8 (1.3)	1.9 (1.1)	0.1 (1.2)	0.85	1.43 (0.196)	1.46 (0.149)	0.03 (0.45)	0.24	−0.07 (0.75)	0.65
sAlb, g/L	38.15 (1.39)	41.08 (0.738)	2.92 (1.62)	0.181	40.94 (0.41)	40.58 (1.31)	−0.36 (1.41)	0.113	−3.28 (0.21)	0.49
Renal Function Test										
sCr, μmol/L	59.15 (2.71)	56.38 (3.48)	−2.76 (3.4)	0.166	58.03 (1.7)	56.77 (1.57)	−1.26 (1.18)	0.296	1.5 (2.22)	0.64
BUN, mmol/L	4.41 (0.33)	4.64 (0.40)	0.23 (0.83)	0.38	4.03 (0.2)	4.42 (0.32)	0.39 (0.45)	0.066	0.16 (0.38)	0.341

^a^ Data are expressed as the mean (SD). ^b^ Mean difference before and after 12 weeks of intake (12 weeks − baseline) ^c^ Within-group differences in the parameters before and after intervention were compared using a paired *t*-test (parametric) and a Wilcoxon Signed Ranks test (nonparametric). Δ ^d^ Mean difference between groups (NRE vs. Placebo) at 12 weeks in the parameters was assessed by independent samples *t*-tests (parametric and equal variances), Welch’s *t*-tests (unequal variances), and Mann–Whitney U tests (nonparametric).

## Data Availability

All data generated from the study is available within this paper and in Appendix A. Additional data that does not fall under ethical restrictions due to human subject involvement can be made available upon request to the corresponding author (S.S.).

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
