# Peer review of "Antiobesity and Hypolipidemic Potential of Nitraria retusa Extract in Overweight/Obese Women: A Randomized, Double-Blind, Placebo-Controlled Pilot Study"

_nutrients, 2024, doi:10.3390/nu16020317_

Round 1

Reviewer 1 Report

Comments and Suggestions for Authors

Reviewer comments MDPI_nutrients

The authors of this manuscript aimed to evaluate anti-obesogenic properties of Nitraria retusa (NRE) extract in a clinical Pilot Study. The authors gave a strong introduction concerning the prevalence of obesity worldwide and associated risks. The demonstrated the need for this type of study, which was to investigate the effects of a natural product as anti-obesity products. The also demonstrated knowledge of the compounds found in NRE that are likely to biological activity. This Pilot study was well designed and took into consideration a wide variety of endpoints to be measured to evaluate changes in weight, lipids, kidney and liver function as safety measures, and other anthropometric indices. The authors found that NRE did reduced body weight of the subjects consuming NRE compared to placebo and this is supported by the tables and figures. The manuscript was very interesting and well written and I enjoyed reviewing it.

1) In section 2.4 Study Protocol, the authors need to include a little more information regarding the numbers of subjects enrolled, exclusions, etc. so that the numbers add up in that section. This information is found in figure one but didn’t appear until later in the manuscript, so please refer to the figure as well.

2) Paragraph starting line 89 was confusing as written. Please rewrite for clarity.

3) Line 53 change “…(WHO) “The” to lowercase “the”.

3) Line 97 “…effects of NRE “is” should be replaced with “are”.

4) Line 103 “…3T3-L1 cells.” Is lacking a citation for that information.

5) Check the document for spaces added in the middle of words, and inside or missing before parenthesis and %.

6) Lines 402-406 the paper would benefit by showing the increased benefit by age in the form of a figure, or %. There was no figure or table referred.

7) Line 295 formatting is required to left align.

Reviewer 2 Report

Comments and Suggestions for Authors

This is a very intersting paper

I have minor comments

1) between effect should not report pre and post but the delta changes only that it is the effect of intervention minus placebo and their p value of interaction. please modify

2) please display all bars with the same colors (only differentiate the colors of intervention and placebo)

3) data analysis in methods should be amplified. please include the normalized data etc.

4) regarding the effect of intervention, please correlated the changes into the intervention in order to define which factors are moving together

5) please define which predictors at baseline could change the delta changes, it is very important define which baseline characteristics could affact the over time changes of the main primary and secondary outcomes

6) into the introduction please cite the following paper, showing that also bergamot shows similiar effect on anti obesity due to the polyphenols content: paper to cite  , 2019. "Efficacy of bergamot: From anti‐inflammatory and anti‐oxidative mechanisms to clinical applications as preventive agent for cardiovascular morbidity, skin diseases"Food Science & Nutrition7(2), pp.369-384.

Reviewer 3 Report

Comments and Suggestions for Authors

The manuscript, “Anti-obesity and Hypolipidemic Potential of Nitraria retusa Extract in overweight/obese Women: A Randomized, Double-Blind, Placebo- Controlled Pilot Study, is informative for anti-obese effects of NRE. However, I'd like to hear a thoughtful answer to my queries below.

1)      Abstract

(1) Wrong statement; The results of statistical processing on the significance   of the changed values (differences; ) between two groups must be specified in the “abstract”. In the case of blood lipids, only change of TG was significantly different, but change of HDLc was not significantly different. (HDLc was significant in preliminary studies; Ref 36)       

2)      Introduction

(1)   When introducing previous research on the funtionality of Nitraria retusa, simply assuming that effective marker substances are bioactive phytochemicals is insufficient as a momorivation for conducting human intervention studies. At least, it is persuasive to introduce NRE’ bioactive substance or preliminary research that have anti-obesity function of NRE.     

3)      Treatment materials

(1)   There is doubt that the 2021 March-harvested NRE used in the preliminary study (Ref 36, Published in Nutrients) and the 2021 April-harvested NRE used in this study contain exactly the same compounds, 20mg flavonoids and 16ug isorhamnetin. The marker materials in NRE were exactly the same, so why was it harvested again at one-month intervals instead of using the same NRE?

(2)   Isorhamnetin is also a type of flavonoid, so if it is not included in 20mg of flavonoids, what are the active compounds in the 20mg flavonoids which were analyzed by HPLC other than Isorhamnetin?

4)      Results

(1)   The results in Figure 2-4 are a repetition of the tables’ results. Since the results can be sufficiently displayed in the tables, it would be better to delete all figures or re-show as figures only the expected values that are significant or that fit the hypothesis of this study.

(2)   In a preliminary study (Ref 36), a high dose of NRE (same substance as in this study) had no effects at all on hematological parameters, liver & renal functions. Why was the same analysis retried in the experiment of this study?

5)      Discussion

(1)   An explanation of the dropout rate of the NRE treatment or the Placebo groups is needed, and in particular, there is a need to discuss why the dropout rate of the Placebo group is higher than that of the NRE treatment group.

(2)   In a preliminary study (Ref 36), the same substance in NRE was effective only on blood HDLc, but in this study, it was effective only on blood TG. What is the difference? My suggestion is that it is better dyslipidemia , which was shown both low HDLc & high TG, may be good marker for NRE.   

(3)   Besides 20mg flavonoids and 16ug isorhamnetin, what other flavonoids do you think influenced the results of this study? Rather than discussing other anti-obesity studies on numerous flavonoids whose significant components included in NRE have not been confirmed, it would be better to focus on Isorhamnetin and explain its mechanism. Also, since Orlistat was not used as a positive control, comparative review would be meaningless.

Comments on the Quality of English Language

.

Round 2

Reviewer 3 Report

Comments and Suggestions for Authors

It has been revised enough to understand some of doubts according to the reviewer's suggestions and has been made available for publication.